# GENERALIZATION TO OUT-OF-DISTRIBUTION TRANS-FORMATIONS

## ABSTRACT

Humans understand a set of canonical geometric transformations (such as translation, rotation and scaling) that support generalization by being untethered to any specific object. We explored inductive biases that allowed artificial neural networks to learn these transformations in pixel space in a way that could generalize out-of-distribution (OOD). Unsurprisingly, we found that convolution and high training diversity were important contributing factors to OOD generalization of translation to untrained shapes, sizes, time-points and locations, however these weren't sufficient for rotation and scaling. To remedy this we show that two more principle components are needed 1) iterative training where outputs are fed back as inputs 2) applying convolutions after conversion to log-polar space. We propose POLARAE, an autoencoder operating in log-polar space which exploits all the four components and outperforms standard convolutional autoencoders and variational autoencoders trained iteratively with high diversity wrt OOD generalization of rotation and scaling transformations.

## 1  INTRODUCTION

Humans have a unique ability to generalize beyond the scope of prior experience (Chollet, 2019; Lake et al., 2017; Marcus, 2001), while artificial agents struggle to apply knowledge to distributions outside the convex hull of their training data (Santoro et al., 2018; Lake & Baroni, 2018). One way humans seem to achieve such generalization is by learning a set of primitive abstract structures, like the 1D ordinal scale (Summerfield et al., 2020) and grid-like representations (Hafting et al., 2005). These structures can also be thought of as symmetry functions: transformations that are in some way invariant to the specific value of their arguments. As a concrete example, we can imagine moving any object around in space, regardless of its shape, color or size. A fundamental question is how can such symmetry functions be learned?

We hypothesize that during development, humans learn a set of canonical transformations - e.g. the translation, rotation and change in size of objects - that are grounded in the sensorimotor system (Barsalou, 2008), and learned as a consequence of predicting the sensory results of primitive actions (Battaglia et al., 2013). The abilities to translate, rotate or scale arbitrary objects then become our first abstract affordances (Gibson, 2014). Indeed, infants that spend more time playing with blocks are better at abstract mental rotation tasks (Schwarzer et al., 2013).

Our aim is to model part of this process by presenting artificial neural networks (LeCun et al., 1995) with 2-dimensional shapes, and training them to predict the effect of translation, rotation or scaling of that shape in pixel space. This is analogous to predicting tactile or visual signals resulting from a simple movement or saccade (Rao & Ballard, 1999; Wolpert et al., 1995). We do not explicitly model motor actions, but rather transform the image offline, and feed the result back to the model. We then test the extent to which such predictions generalize OOD along dimensions such as shape, size, location and time. Evidence of OOD generalization would suggest that the network has learned a symmetry function.

We assume that, in order to learn symmetry functions, we must introduce principled inductive biases. The first is convolution. Second, to constrain the network to learn a primitive function that can apply to any shape, we assume it requires exposure to a sufficiently diverse set of examples. We therefore operationalize and vary 'diversity' as the number of distinct shapes present in the training set. As the translational invariance built into convolution is well aligned with the task of translation

unsurprisingly, a fully convolutional autoencoder with high training set diversity acheives near perfect OOD generalization for translation to unseen shapes in larger grids and new locations (Figure 1). However the two components weren't sufficient for OOD generalization of rotation (Figure 2) and scaling. To remedy this we introduce two more principle components. First is the effect of 'iteration' during training (output fed back as input iteratively), based on the idea that sequential applications of the same transformation should maintain the identity of an object (i.e. object permanence (Piaget, 2006)). Secondly, motivated by the strong OOD generalization of translation as depicted in Figure 1, and the fact that translation in log-polar space is equivalent to rotation and scaling in cartesian space (Figure 1 of (Esteves et al., 2017; Tootell et al., 1982)), we hypothesize that performing convolutions in log polar space would help in the OOD generalization of rotation and scaling.

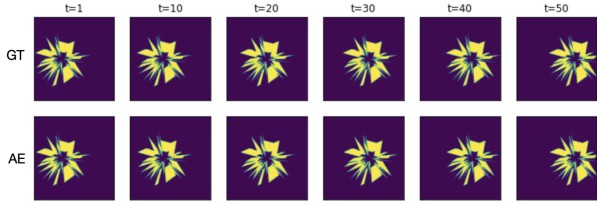

Figure 1: Translation OOD (larger, hollow, shapes with 50 vertices on a 256x256 grid (trained on 64x64 grid). Ground truth (GT) and autoencoder (AE) predictions displayed.

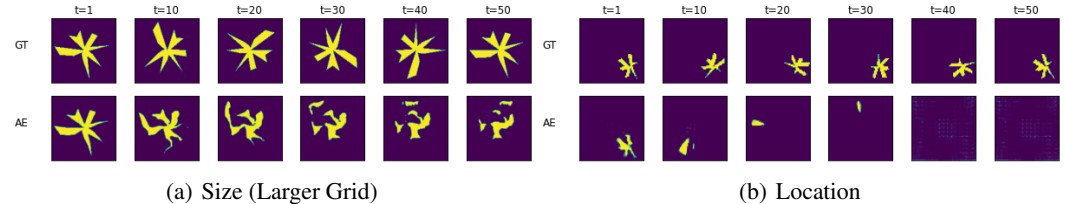

(a) Size (Larger Grid)         (b) Location

Figure 2: Comparison of OOD rotation between AE trained with high diversity and GT across multiple times.

We summarize the following contributions:

- We build a flexible data generator that can produce and transform a large variety of simple but structured images - irregular polygons with variable size, location, and complexity.
- We show that iteration has the ability to conserve the shape far past the time horizon seen during training and is necessary for better OOD generalization of rotation and scaling transformations along with convolution and high diversity.
- We find an interesting tradeoff between diversity and iteration where each could partially make up for less of the other to produce better OOD generalization capabilities.
- We novelly propose POLARAE, a fully convolutional autoencoder build using concepts from polar transformer networks (Esteves et al., 2017) which exploits all the four components demonstrating better OOD generalization performance than a standard autoencoder and a variational autoencoder in cartesian space.

## 2 RELATED WORK

### 2.1 PSYCHOLOGY

Humans can perform mental transformations OOD (i.e. on unseen shapes). For example, humans can mentally translate or rotate shapes at a steady rate (Shepard & Metzler, 1971) or scale abstract distances (Trope & Liberman, 2010). There are also known mechanisms that might make these

transformations general. For example, there is a known impact of diversity on the generalization of properties in psychology, namely, the diversity effect (Osherson et al., 1990). Iteration is also a plausible mechanism in humans - psychological data for mental rotation suggest that we transform objects iteratively, since larger angles of rotation elicit longer reaction times (Shepard & Metzler, 1971). There is also evidence that discrete temporal context updating by recurrent thalamocortical loops serves predictive learning in the brain (O'Reilly et al., 2014). Basically, iterated operations achieve generality since they are time-invariant. Finally, humans have built-in architectural transforms, such as the transformation of retinal images into a log-polar coordinate system (Maiello et al., 2020).

## 2.2 MACHINE LEARNING

There have been several works that demonstrate that high diversity is necessary for OOD generalization. Xu et al. (2020) provided theoretical guarantees that ReLU MLP networks extrapolate well with linear functions and sufficient diversity in their training set . Sufficient diversity of input is also required for systematic generalization of neural network based reinforcement learning agents (Hill et al., 2019). Madan et al. (2021) showed that data diversity improves OOD category-viewpoint classification but didn't focus on the generative aspect of it using CNNs. There have also been works collecting large scale controlled synthetic datasets (Borji et al., 2016; Gross et al., 2010; Qiu et al., 2017) to facilitate learning invariance to different types of transformations in deep neural networks. Following a similar motivation, we want CNNs to be able to learn such transformations and to be able to generate them on OOD examples. The idea of iterative training has also been explored in this domain. Using an iterative training technique similar to ours generative adversarial networks learned 3D rotations (but not scaling) (Galama & Mensink, 2019). However there were limited assessment of how different amounts of iteration during training impacted extrapolation in time. Generally, predicting past the number of iterations seen during training has required building in a conservation law of some sort (Cranmer et al., 2020; Greydanus et al., 2019). Kumar et al. (2020) showed that gradual self training at each iteration helps in domain adaptation. In a different context, Kuchaiev & Ginsburg (2017) proposed iterative output re-feeding to train deep autoencoders for improved collaborative filtering. Recently, the idea that iteration can achieve OOD generalization has been put forth in the context of problem solving (Schwarzschild et al., 2021). Kim et al. (2020) explored the effectiveness of log-polar space in achieving rotation invariance but was limited to classification and didn't study other types of transformations like scaling.

## 3 METHOD

### 3.1 DATA GENERATOR

All training stimuli were shapes contained within a 64x64 pixel grid space. We constructed irregular N-sided polygonal shapes by first sampling N angular distances between 0 and $2\pi$, and then sampling a radial distance from a centroid ( $x_{center}$, $y_{center}$) at each of these angles uniformly between 0 and a scale parameter $r$. This produced a set of vertices; pixels within the convexity of the vertices were set to 1, and pixels outside were set to 0. There was also the option to produce 'hollow' shapes such that an interior cut-out of the shape was set to 0, in order to produce a test-set of shapes with different distribution from the training set. This data generator was therefore capable of producing a combinatorically large set of possible shapes. Our procedure for shape generation is most similar to Attneave forms (Attneave & Arnoult, 1956) and also bears relation to the method of Fourier descriptors (Zhang & Lu, 2005), but was selected due to its computational speed and interpretable manipulations of shape parameters. Each shape was used as the input to a neural network, and was transformed in one of the following ways to generate the target for training: for translation, shift 2 pixels to the right; for rotation, rotate $\frac{\pi}{25}$ radians clockwise; for scaling, increase radial length of the vertices by 0.1 (Figure 3). These transformations were hard-coded but meant to represent a set of innate primitive actions.

### 3.2 MODEL ARCHITECTURE

We first describe the three baseline models followed by the proposed model:

Figure 3: Transformations of shapes produced from the data generator.

The first model was a fully convolutional autoencoder. The encoder consisted of 3 convolutional layers, the first with 16 3x3 kernels, (stride=2, padding=1), the second with 32 3x3 kernels (stride=2, padding=1), and the last with 64 7x7 kernels (stride=1). The decoder had 3 layers (padding=1, output padding=1) that inverted these operations with transposed convolution layers that were mirror images of the encoder layers. This produced an output with the same dimensions as the input image. All layers were followed by rectified linear (ReLU) activations. Being fully convolutional, it could accept any input grid size. All weights were initialized using Xavier uniform initialization. We use AE to denote this model.

The second model was a fully convolutional variational autoencoder (Kingma & Welling, 2013). The encoder and decoder architectures were similar to AE. We use VAE to denote this model.

The third model was a $\beta$-VAE (Higgins et al., 2016). We used $\beta = 4$ since that learnt a good disentangled representation of the data generative factors in the original work.

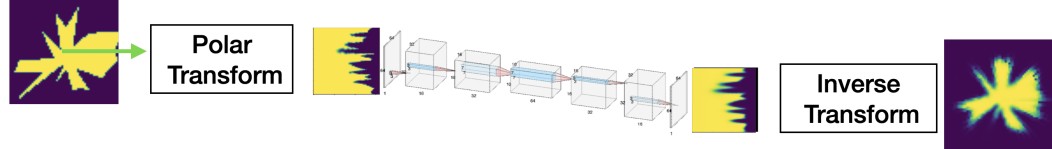

Figure 4: POLARAE architecture: The input image is passed through a polar transform module which consists of polar origin predictor followed by conversion to log-polar space. The obtained polar representation is passed through a fully convolutional autoencoder and finally the output is passed through an inverse transform module to convert the image back to cartesian coordinate space.

The proposed model was a combination of polar transformer networks and the fully convolutional autoencoder described above. More specifically, given an input image the polar origin predictor computed a single channel feature map using a fully convolutional network and the centroid of the heatmap was used as the polar origin. The polar transformer module converted the image to log polar coordinates using the predicted polar origin. The transformed image was then fed to AE. Since the rotation of the input image resulted in vertical shift in log polar space wrapping at the boundary, we used wrap around padding on the vertical dimension and zero padding in the horizontal dimension before applying convolutions for the first two layers of the encoder. An inverse polar transformer module was used to convert the output of the AE from log polar space to the original image space. The inverse polar transformer module used the sample points of the original input image used to compute the log-polar transform and mean approximation since a single point in the original image contributed to multiple points in the log-polar space to convert the image back to cartesian coordinates. Since this is effectively an autoencoder operating in the log-polar space we use POLARAE to denote this model. Figure 4 shows the architecture.

## 3.3 TRAINING AND TESTING

All networks were trained by providing randomly generated shapes with $N \in [20, 21]$, and back-propagating the mean squared error (MSE) loss between the output and the appropriately transformed target shapes. Weight updates were performed with the ADAM optimizer, using a learning rate of 2.5e-4, weight decay of 1e-5, and batch size ($BS$) of 32. Networks were all trained for $Nsteps$=100000. We trained separate networks for translation, rotation, and scaling, but the training procedure and the variation in diversity and iteration as described next was common to all networks.

For rotation, inputs were centered filled shapes with $r \in [10,10]$, $x_{center}$ = 32 , $y_{center}$ = 32, and targets were input shapes rotated clockwise by $\frac{\pi k}{25}$ radians, where $k$ is the numbers of iterations. Note $k = 50$ would amount to full rotation and we evaluate our networks till that. These set of networks were tested for OOD generalization to hollow shapes and larger shapes in larger grids. The set of networks which were tested for OOD generalization of rotation to new locations were trained on filled shapes in the upper left region with $r \in [5,7]$, $x_{center}, y_{center} \in [14,18]$ . Finally the set of networks which were tested on larger shapes on the same grid size were trained on centered filled shapes with $r \in [5,7]$. For scaling, inputs were centered filled shapes with $r \in [5,7]$, and targets were input shapes with the radial distance of the vertices increased by $0.1k$. These set of networks were tested for OOD generalization to hollow shapes and larger shapes in larger grids. To test OOD generalization at new locations, networks were trained with $r \in [5,7]$, $x_{center}, y_{center} \in [22,26]$. To generate the targets for this case the radial distances were increased by $0.08k$.

To vary diversity ($D$), we trained a separate set of networks for each of 100, 1000, 10000 and $inf$ items in the training set ($inf$ involved randomly generating new images on the fly every step, to approximate an infinite diversity of shapes). To vary iteration, we introduced a training variant (Algorithm 1 in Appendix) that treated the network as an iterated function, based on the principle that $f(n_{t+1}) = f(f(n_t))$, where $f$ is a rotation or a scaling function approximated by model $M$, and $n_t$ is a shape after $t$ transformations. For each input, the final output was generated by $k$ applications of $M$ (Line #5-7 in Algorithm 1). The final output was compared with the target image (initial image transformed $k$ times, for example, $\frac{k\pi}{25}$ clockwise rotations). The loss and the gradients were calculated on the final output and then used to perform a single weight update (Line #8-10). For each training step, the integer $k$ was sampled uniformly between 1 and $I_t$ (Line #3). To assess the effect of varying the amount of iteration we trained seperate set of models with $I_t = 1, 2, 9$. These values were mainly chosen to study the effect of no iteration ($I_t = 1$), just one step of iterative training ($I_t = 2$) and sufficient number of steps ($I_t = 9$) which is still quite less than the number of iterations used during testing as mentioned later. While this procedure resembles recurrent neural networks like the LSTM (Hochreiter & Schmidhuber, 1997), it differs in that it accepts only a single input (whereas recurrent networks usually accept sequences) and it only propagates the output, rather than a hidden network state, so it is memory-less.

At test time, the networks were presented with new images which it hadn't seen before, and then repeatedly applied (outputs passed back as inputs) to assess the correspondence between each successive transformation and the correct shape at that time point. We used the intersection over union (IOU) metric to assess test performance at time = 1 and time = 50 (temporal extrapolation since it reflects transformations far past the horizon used during training) . We used a selection of test datasets to assess OOD generalization along different dimensions. For the dimension of Shape, we tested on shapes that had a hollow center since training shapes were all filled. For Location, we tested in the lower right region with $x_{center}, y_{center} \in [46,50]$ for rotation and $x_{center}, y_{center} \in [38,42]$ for scaling transformation. For Size (Larger grid), we doubled the grid-size from 64x64 to 128x128 and doubled the radius of the shapes so they were much larger than shapes used for training. For Size (Same grid), the radius of the shapes were increased from [5,7] to 10 on 64x64 grid. Each test data set contained 500 shapes.

## 4 RESULTS

We provide mean IOU scores along with standard deviation across five runs for each instance.

### 4.1 ROTATION

To demonstrate the effect of iteration we first focused on only extrapolation in time; since networks were trained on a small number (1-9) of iterations, repeatedly transforming stimuli much longer than that at test time (up to 50 times) would reflect extrapolation. As well, due to the combinatorial size of the shape space, I.I.D. generalization rather than memorization was still necessary to succeed at this task. As seen from Figure 5(a), the AE trained with ($D = inf, I_t = 9$) was able to conserve the shape even till $t = 50$ and also achieved highest IOU (Figure 5(b)). This indicates that iteration tightens the error tolerance of each network output to achieve greater stability in time. There is also an interesting tradeoff between diversity and iteration shown in Figure 5(b) apparently for $t = 50$. For lower diversity and higher iteration ($D = 100, I_t = 9$), the mean IOU score is very similar

(slightly better) to the case with higher diversity, lower iteration ($D = inf, I_t = 1$). These iterative networks were also more stable in time than networks trained to rotate large angles in a single pass (Figure 6), so iterative networks rotate at smaller increments while maintaining greater accuracy over time. However even with high diversity and iteration ($D = inf, I_t = 9$), AE failed to rotate larger shapes in larger grids (Figure 7(a)) or at new locations (Figure 7(b)), suggesting that only the three components aren't sufficient.

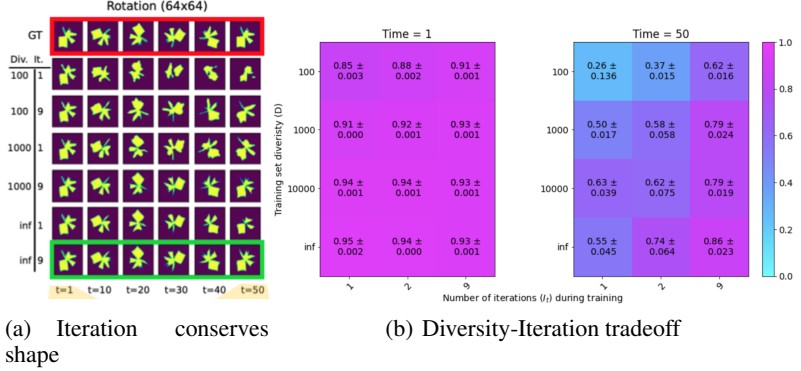

(a) Iteration conserves shape

(b) Diversity-Iteration tradeoff

Figure 5: Effect of diversity,iteration on IID rotation quality for AE.

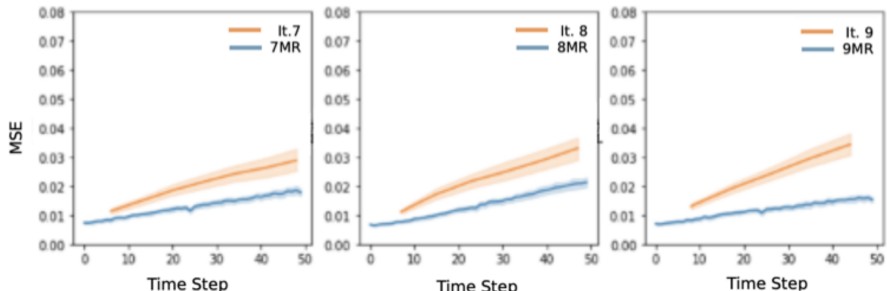

Figure 6: Comparison of iteratively trained rotation networks ($I_t$= 7, 8, 9) with networks that were trained to rotate a specified angular distance in a single forward pass. For example, if the network $I_t = 9$ experienced between 1 and 9 iterative passes during training, then the comparison network was trained to rotate an object by 9 minimal rotations (9MR or $\frac{9\pi}{25}$ radians) in a single pass. For higher values of iterations/MR, the iteratively trained networks have improved performance over time, as indicated by MSE, despite rotating shapes using many more steps.

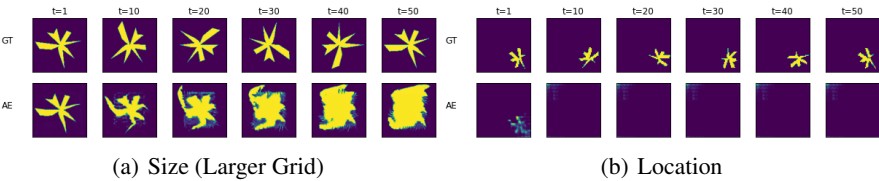

(a) Size (Larger Grid)

(b) Location

Figure 7: Comparison of OOD rotation quality between ground truth (GT), AE across time.

From Table 1 we can see that POLARAE achieves about 28% and 44% improvement in mean IOU score for OOD generalization to larger shapes in larger grid and location respectively compared to the second best performing model VAE. Figure 8(a), 8(b) demostrates that POLARAE can rotate larger shapes in larger grids and new locations respectively, whereas VAE distorts the shapes, or cannot detect it and is not able to rotate in time.

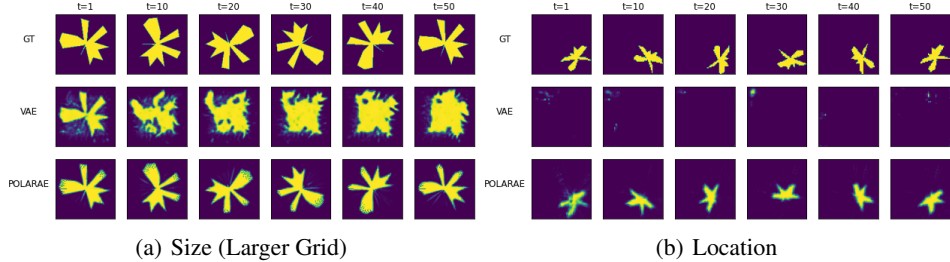

(a) Size (Larger Grid)  (b) Location

Figure 8: Comparison of OOD rotation quality between ground truth (GT), VAE, POLARAE across multiple times.

Figure 9 shows the effect of diversity and iteration for POLARAE at OOD generalization to larger shapes in larger grids (9(a)) and unseen locations (9(b)). As indicated by the colour gradient for the case of $t = 50$, best performance was obtained by maximizing both diversity and iteration during training, and as pointed out earlier lesser diversity can be compensated by using higher iteration.

For OOD generalization to hollow shapes and larger shapes in the same grid there were no major qualitative differences in the rotation over time (Figure 17 in Appendix).

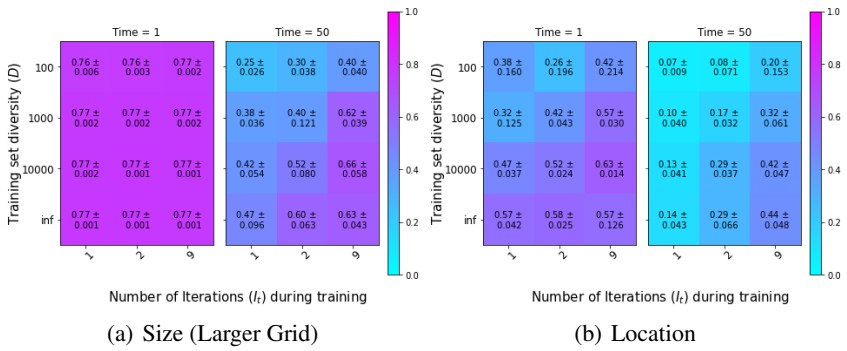

(a) Size (Larger Grid)  (b) Location

Figure 9: Effect of diversity vs iteration for POLARAE on OOD rotation

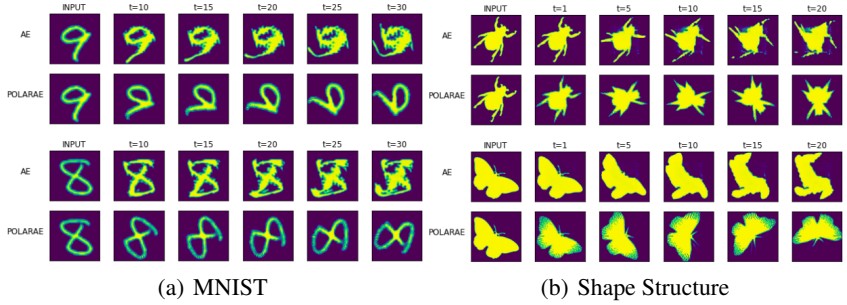

(a) MNIST  (b) Shape Structure

Figure 10: Comparison of OOD rotation quality between AE, POLARAE across time.

We also experimented with two more 2D datasets, MNIST (LeCun et al., 1998) and Shape Structure (Carlier et al., 2016). For MNIST we trained the networks iteratively to rotate digits 0-5 on a 64x64 grid and tested on digits 6-9 on a larger 128x128 grid. As we can see from Figure 10(a) POLARAE was able to succesfully rotate novel shapes presented in a larger grid with time whereas AE failed. For the Shape Structure dataset we just evaluated the performance of the networks on novel shapes with time. (Figure 10(b)).

Table 1: IOU scores of trained models tested OOD for rotation at time-points 1 and 50.

| Model | Shape | | Size (Same grid) | | Size (Larger grid) | | Location | |
|---|---|---|---|---|---|---|---|---|
| | $t = 1$ | $t = 50$ | $t = 1$ | $t = 50$ | $t = 1$ | $t = 50$ | $t = 1$ | $t = 50$ |
| AE | $\mathbf{0.86}^{\pm 0.07}$ | $0.57^{\pm 0.04}$ | $\mathbf{0.89}^{\pm 0.03}$ | $0.62^{\pm 0.06}$ | $0.68^{\pm 0.07}$ | $0.35^{\pm 0.07}$ | $0.28^{\pm 0.08}$ | $0.00^{\pm 0.00}$ |
| VAE | $0.72^{\pm 0.00}$ | $0.38^{\pm 0.01}$ | $0.76^{\pm 0.00}$ | $0.36^{\pm 0.03}$ | $0.71^{\pm 0.00}$ | $0.38^{\pm 0.01}$ | $0.01^{\pm 0.00}$ | $0.00^{\pm 0.00}$ |
| $\beta$-VAE | $0.59^{\pm 0.00}$ | $0.24^{\pm 0.04}$ | $0.65^{\pm 0.01}$ | $0.28^{\pm 0.02}$ | $0.69^{\pm 0.01}$ | $0.36^{\pm 0.01}$ | $0.08^{\pm 0.02}$ | $0.00^{\pm 0.00}$ |
| POLARAE | $0.76^{\pm 0.01}$ | $\mathbf{0.61}^{\pm 0.03}$ | $0.88^{\pm 0.00}$ | $\mathbf{0.66}^{\pm 0.05}$ | $\mathbf{0.77}^{\pm 0.00}$ | $\mathbf{0.66}^{\pm 0.06}$ | $\mathbf{0.63}^{\pm 0.01}$ | $\mathbf{0.44}^{\pm 0.05}$ |

## 4.2 SCALING

For scaling transformation too, the best performing AE at $t = 50$ was trained with $(D = inf, I_t = 9)$, and the diversity-iteration followed a similar pattern as in Fig. 5(b). However even with high diversity and iteration, AE failed to scale larger shapes in larger grids (Figure 13(a) in Appendix) or at new locations (Figure 13(b) in Appendix).

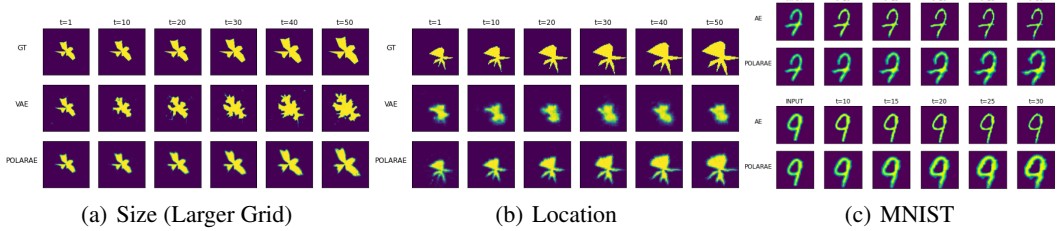

(a) Size (Larger Grid)      (b) Location      (c) MNIST

Figure 11: Comparison of OOD scaling quality between ground truth (GT), VAE, AE, POLARAE across multiple times.

From Table 2 we can see that POLARAE achieves 27% and 29% improvement in mean IOU score for OOD generalization to larger shapes in larger grid and location respectively compared to the second best performing model VAE. Figure 11(a), 11(b) demostrates that POLARAE can scale larger shapes in larger grids and new locations respectively, whereas VAE distorts the shapes without actually scaling it with time. POLARAE can also scale unseen MNIST digits presented on a larger grid in time unlike AE (Figure 11(c)).

Figure 14 in Appendix shows the trade-off between diversity and iteration of POLARAE. As indicated by the colour gradient for the case of $t = 50$, best performance here too was obtained for high diveristy( $D = inf$ ) and high iteration( $I_t = 9$ ). Here too for lower diversity and higher iteration $(D = 1000, I_t = 9)$, the mean IOU score is very similar to the case with higher diversity, lower iteration $(D = inf, I_t = 1)$.

Table 2: IOU scores of trained models tested OOD for scaling at time-points 1 and 50.

| Model | Shape | | Size (Same grid) | | Size (Larger grid) | | Location | |
|---|---|---|---|---|---|---|---|---|
| | $t = 1$ | $t = 50$ | $t = 1$ | $t = 50$ | $t = 1$ | $t = 50$ | $t = 1$ | $t = 50$ |
| AE | $\mathbf{0.87}^{\pm 0.01}$ | $0.44^{\pm 0.01}$ | $\mathbf{0.94}^{\pm 0.01}$ | $\mathbf{0.75}^{\pm 0.00}$ | $\mathbf{0.97}^{\pm 0.01}$ | $0.42^{\pm 0.01}$ | $0.79^{\pm 0.05}$ | $0.23^{\pm 0.03}$ |
| VAE | $0.67^{\pm 0.00}$ | $0.22^{\pm 0.02}$ | $0.85^{\pm 0.00}$ | $0.41^{0.03}$ | $0.91^{0.00}$ | $0.49^{\pm 0.02}$ | $0.62^{\pm 0.01}$ | $0.28^{\pm 0.03}$ |
| $\beta$-VAE | $0.55^{\pm 0.01}$ | $0.17^{\pm 0.04}$ | $0.72^{\pm 0.01}$ | $0.28^{\pm 0.06}$ | $0.55^{\pm 0.06}$ | $0.25^{\pm 0.03}$ | $0.30^{\pm 0.05}$ | $0.09^{\pm 0.01}$ |
| POLARAE | $0.81^{\pm 0.00}$ | $\mathbf{0.46}^{\pm 0.00}$ | $0.89^{\pm 0.00}$ | $0.71^{\pm 0.01}$ | $0.95^{\pm 0.00}$ | $\mathbf{0.76}^{\pm 0.01}$ | $\mathbf{0.82}^{\pm 0.01}$ | $\mathbf{0.57}^{\pm 0.03}$ |

## 4.3 EXTENSION TO 3D SHAPES

We extended the models to perform 3D object rotation from voxel occupancy grids. We used ModelNet40 (Wu et al., 2015), where objects were rotated around $z$ direction. We iteratively trained to

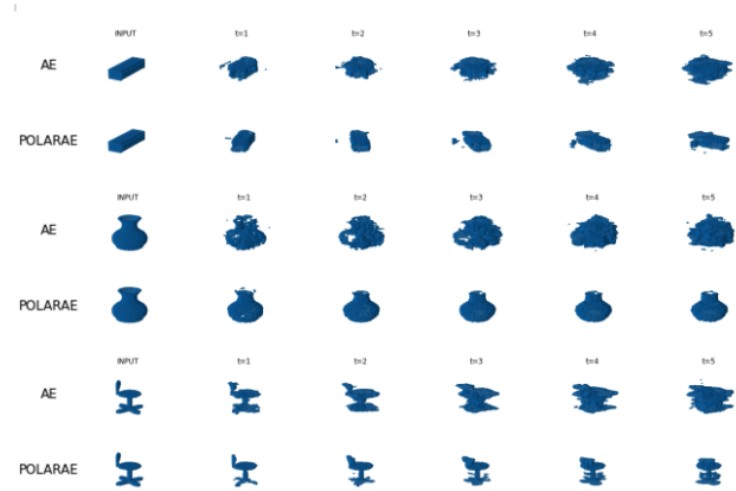

Figure 12: Comparison of OOD rotation quality of 3D shapes across time.

rotate 30 object categories on 30x30x30 voxel grids and tested on the heldout 10 categories presented in larger 50x50x50 grids. We used techniques from Esteves et al. (2017) to extend POLARAE to 3D, predicting an axis and using it as origin to compute cylindrical coordinates (channel-wise polar transformation). For all channels, the origin was the same and each channel was a 2D slice of the 3D voxel grid. Similarly the inverse transform module was extended to 3D to convert the output back to voxel grid. For AE we used a fully 3D convolutional autoencoder. As we can see from Figure 12, POLARAE is able to rotate unseen 3D objects with time barring minor distortions whereas AE completely distorts it and is unable to rotate.

## 5 Discussion & Future directions

We tested the ability of standard autoencoders, variational autoencoders in cartesian space and autoencoders in log-polar space to extrapolate learned transformations in pixel space. Unsurprisingly, convolution and high diversity were sufficient for OOD generalization of translation. However, those weren't sufficient for OOD generalization of rotation and scaling. Iterative training helped preserve the shape far past the time-horizon seen during training, but was still not sufficient to rotate larger shapes in larger grids or at new locations with time.

Performing standard convolutions in log-polar space combined with high diversity and high number of iterations during training resulted in stable transformations for larger shapes in larger grids and new locations far past the time horizon seen during training. Here, we propose that conservation of shape emerges from the symmetry implied by training the networks as iterated functions; as Noether's theorem states, for any symmetric action, there is a corresponding conservation law (Noether, 1971).

We also found an interesting interaction between diversity and iteration, where each could partially make up for less of the other to produce OOD generalization capacities. Based on all these results, we suggest that humans may use these strategies synergistically during development in order to learn canonical transformations. Intuitively, a child may optimally learn to predict the sensory effects of these transformations by both transforming many different objects as well as repeatedly transforming the same object. In this way, they are able to abstract canonical transformations away from individual object instances, while maintaining a sense of object permanence.

Finally, it appeared that transforming inputs into an appropriate coordinate space using POLARAE significantly improved OOD generalization to rotation, scaling. We believe that the polar coordinate space was better able to leverage the inductive bias of translational convolution. We hope that these mechanisms inspired from humans can be incorporated into future neural network models to improve generalization to downstream tasks like classification, localization.

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

## A APPENDIX

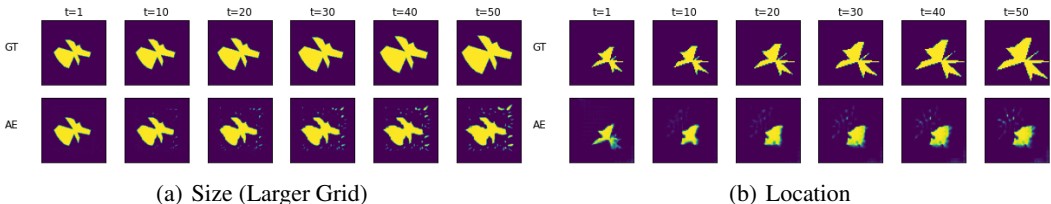

(a) Size (Larger Grid)        (b) Location

Figure 13: Comparison of OOD scaling quality between ground truth (GT), AE across time.

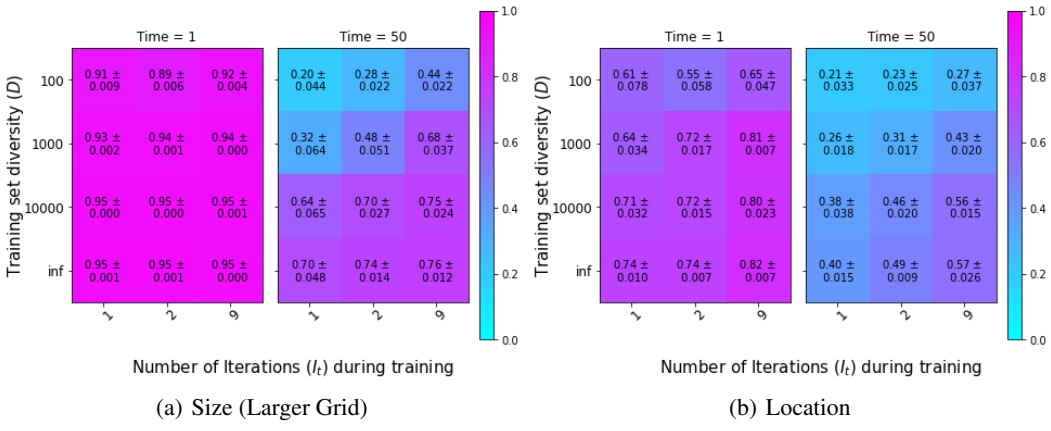

(a) Size (Larger Grid)

(b) Location

Figure 14: Effect of diversity vs iteration for POLARAE on OOD scaling

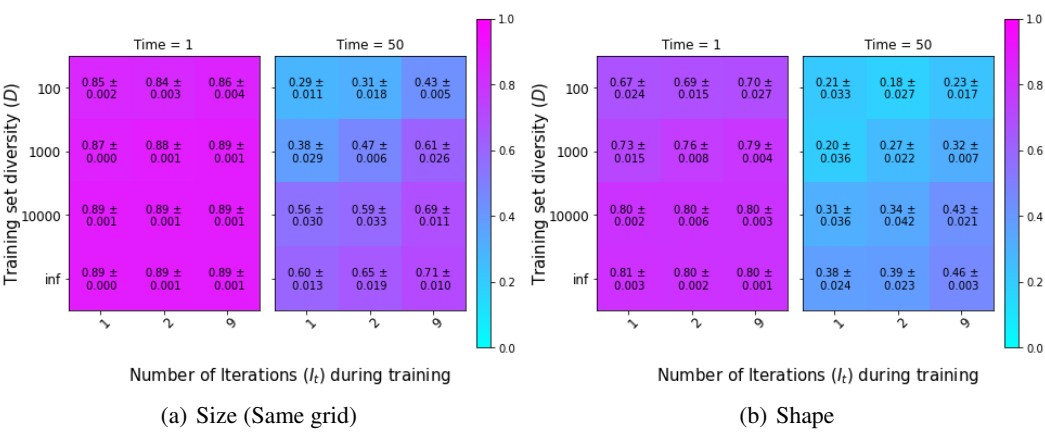

(a) Size (Same grid)

(b) Shape

Figure 15: Effect of diversity vs iteration for POLARAE on OOD scaling

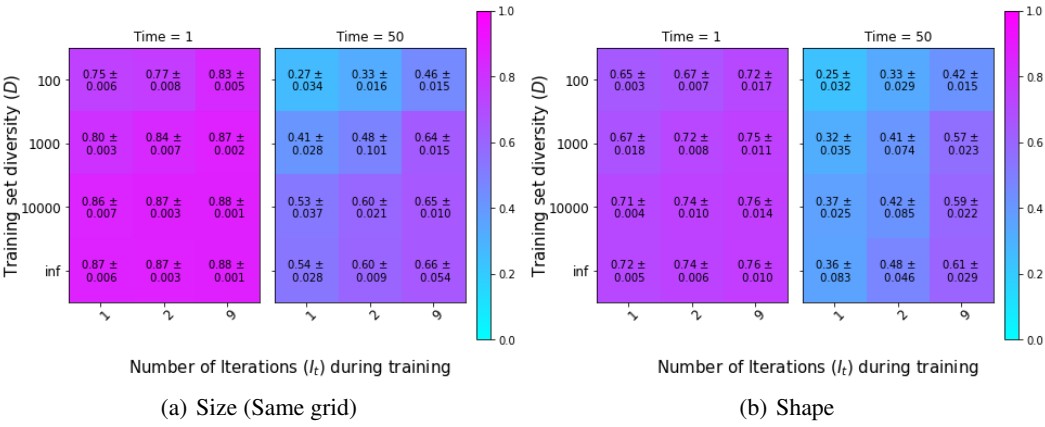

(a) Size (Same grid)

(b) Shape

Figure 16: Effect of diversity vs iteration for POLARAE on OOD rotation

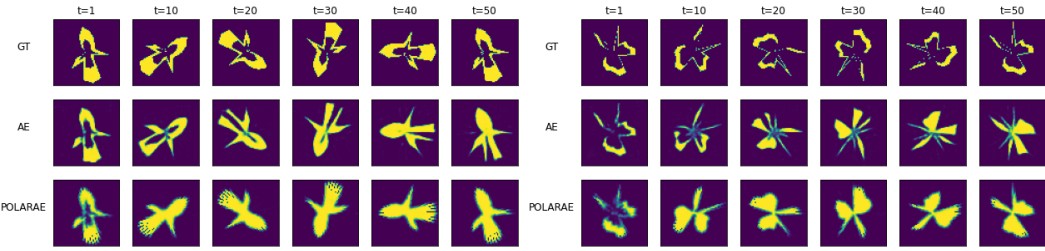

Figure 17: Comparison of OOD rotation on hollow shapes.

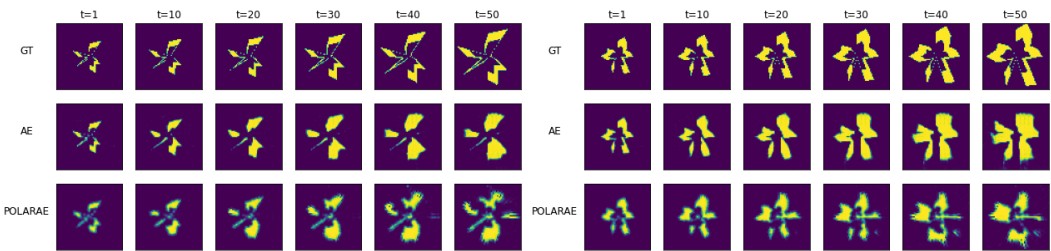

Figure 18: Comparison of OOD scaling on hollow shapes.

---

**Algorithm 1:** Iterative training with $D = inf$

---

**1** **Initialize:** Model $M$ with parameter $\theta$;

**2** **for** *step s=1 to Nsteps* **do**

**3**     $k \leftarrow$ random integer drawn uniformly from $[1, I_t]$;

**4**     $input, target \leftarrow$ data_generator$(r, x_{center}, y_{center}, N, k, BS)$;

**5**     **for** *iteration iter=1 to k* **do**

**6**        $output \leftarrow M(input)$

**7**        $input \leftarrow output$

**8**     $loss \leftarrow$ MSELoss$(output, target)$;

**9**     $loss$.backward();

**10**     Update parameter $\theta$ ;

---

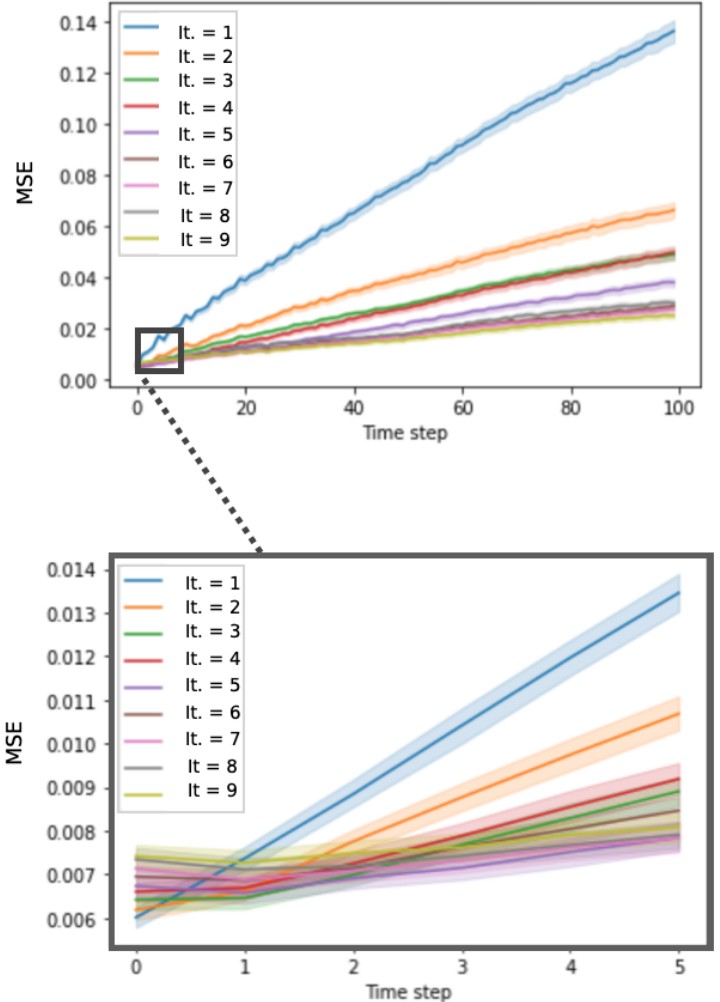

Figure 19: Top: Mean squared error (MSE) on I.I.D. test set at each time step for 9 trained rotation networks. The label 'It. = N' indicates that during training, the number of iterations of the newtork on a given batch was sampled uniformly between 1 and N. Bottom: the same plot blown up to visualize the first 6 time-steps. Of note, networks trained with higher iterations actually have a worse MSE at the first time step, but achieve a much better MSE in the long run. Confidence intervals represent standard error between 3 identically trained networks.

