# OpenReview forum: "Generalization to Out-of-Distribution transformations"
_ICLR.cc/2022/Conference — ICLR 2022 Submitted_

### Official Review · Reviewer_3Ri2 · 2021-10-28

**Correctness:** 1
**Technical Novelty And Significance:** 2
**Empirical Novelty And Significance:** 2
**Recommendation:** 3
**Confidence:** 3

**Main Review:**

The key contribution of this paper is to propose to utilize convolution, high training diversity, iterative training, and log-polar space for improving the generalization ability of AE models for OOD settings. The authors demonstrate its usefulness on a toy task.

Though the authors start the paper with psychology literature and discussions, they fail to really establish connections between their proposed methods and the literature.

For example, the authors write in the paper "...we hypothesize that performing convolutions in log polar space would help in the OOD generalization of rotation and scaling", but they never provide any, not even distantly related, support from psychology. O the other hand, there is no mathematical justification, similar to the paper [https://arxiv.org/abs/1709.0188](https://arxiv.org/abs/1709.01889](https://arxiv.org/abs/1709.01889),)

As for experiments, the authors propose a new data generator to create lots of simple but structured images. In my opinion, this dataset is too simple, and cannot justify the claim of the authors. I don't know why the authors do not use the 3D dataset used in [https://arxiv.org/abs/1709.0188](https://arxiv.org/abs/1709.01889](https://arxiv.org/abs/1709.01889),), which seems more difficult than the one used in this work.

The writing of this paper seems sloppy and needs to be improved significantly. For example in the abstract, the name of the proposed method, POLARAE, is introduced but its meaning has never been properly explained.

Many figures in the paper need to be improved:

- It's difficult to read the numbers in Figure 4
- Figure 6 and Figure 17 are very blurry and should use PDF formats.

**Summary Of The Paper:**

Though the authors start the paper with psychology literature and discussions, they fail to really establish connections between their proposed methods and the literature.

**Summary Of The Review:**

Though it is useful to investigate the OOD transformation on toy tasks first, it is important to scale it up to more realistic settings later. The observations in this paper are not likely to generalize to realistic settings that we really care about.

---

### Official Review · Reviewer_sNFP · 2021-10-30

**Correctness:** 3
**Technical Novelty And Significance:** 2
**Empirical Novelty And Significance:** 2
**Recommendation:** 3
**Confidence:** 3

**Main Review:**

Pros:
1. The evaluation is technically sound.
2. Experiments shows that operating in log-polar space does have in o.o.d generalization for affine transformations.

Cons:
1. Only a very simple synthetic dataset is used for evaluation. I suggest authors to use at least 2 or 3 datasets (such as the Shape dataset) to prove that their method's effectiveness is generalisable to other datasets.
2. Baselines are very limited. Only vanilla CNN and VAE are compared against. There are many more VAE variations that can be compared. For example, AIR [1] model factorises transformations and content by designing an explicit affine-transformation module, while NeRF-VAE [2] uses neural radiant fields.
3. The paper has limited novelty. The proposed model is a VAE in log-polar space. The properties of polar space have already been explored in various papers mentioned by the authors. Recently its effectiveness in achieving rotation invariance has also been explored[3].
4. Many design choices seem arbitrary without proper explanations. For example, why is I_t chosen to be 1,2,9?

Minor:
1. Some figures (figure6) are of very low resolutions.
2. The method can be described more in maths, less in text.
3. Algorithm 1 should be in the main text, not in Appendix. Or if the authors want to put it in Appendix, mention it (say 'Algorithm 1 in Appendix...')

Ref:
1. Eslami, S. M., et al. "Attend, infer, repeat: Fast scene understanding with generative models." Advances in Neural Information Processing Systems 29 (2016): 3225-3233
2. Kosiorek, Adam R., et al. "Nerf-vae: A geometry aware 3d scene generative model." arXiv preprint arXiv:2104.00587 (2021).
3. Kim, Jinpyo, et al. "CyCNN: a rotation invariant CNN using polar mapping and cylindrical convolution layers." arXiv preprint arXiv:2007.10588 (2020)

**Summary Of The Paper:**

The paper proposed POLARAE, an auto-encoder that operates in log-polar space. The paper demonstrated the model, together with iterative learning, is better than simple baselines in o.o.d generalization for a set of affine transformations.

**Summary Of The Review:**

A technically sound paper that has limited novelty. More baselines should be compared against on more datasets.

---

### Official Review · Reviewer_JcBj · 2021-11-02

**Correctness:** 2
**Technical Novelty And Significance:** 1
**Empirical Novelty And Significance:** 2
**Recommendation:** 3
**Confidence:** 4

**Main Review:**

Strengths:

1. The paper proposes interesting new ideas (iteration, log-polar convolutions) to approach a long standing, seminal problem in computer vision.

2. Experiment design is solid, and design decisions take care to ensure a high degree of scientific rigor.

3. Paper is very well written. It is easy to follow and understand.

4. The approach used here---empirical, controlled analysis of generalization is a great tool to better understand and improve generalization performance of computer vision models.

Weaknesses:

1. All experiments are conducted on a very simple toy dataset, despite several more complex datasets existing: The paper suggests the use of this dataset generation pipeline because it provides them complete control over the training and test distributions, interpretable manipulations and computational speed. However, rotational invariance is a long standing problem, and several datasets exist to study it. Especially in 2D. For instance the popular MNIST-Rotation dataset. In fact, if we limit the problem to 2D rotations, it is possible to take almost any image dataset and create 2D variations of it with great computational speed. In fact, there are several datasets that exist to study 3D rotation invariance now, which all allow generating new rotations (albeit at a greater computational cost). In is unclear how well these findings extend to more complex data, which significantly reduces the utility.

2. Use of boundary points by the network: As mentioned in Sec 3.2, the inverse transform module uses the sample points of the original input image. This presents a problem, as implementing this approach on any other dataset would first require segmenting the objects to obtain the boundary control points. Thus, POLARAE can only be used if segmentation masks exist, which limits its applicability. This is also tricky because for some experiments (hollow OOD samples) as this information is the same for in-distribution and OOD samples. If the model relies heavily on control points, it would explain why hollow shapes are not a problem for the model.

3. Overlapping train/test distributions, not truly OOD: Sec 3.3 mentions that fir scaling, input images had r in range (5,7), while test images were created by scaling vertices in increments of 0.1. Polygons for which r was towards the lower end of this spectrum would still be within the training distribution range of (5,7) for multiple iterations of scaling. Thus, the test target is not guaranteed to be OOD. To ensure it, a range of 7+delta would be necessary. Same is true for rotation. A rotated target sample could by itself also be generated by the training procedure itself with a non-zero probability. Thus, the the train and test distributions are not disjoint, and thus overlapping. A similar logic holds true for location as well. Thus the only purely OOD case is of the hollow shapes, but that suffers from its own problem as mentioned above.

4. Baselines: It is unclear what the paper means by "understanding a transformation", which makes baselines tricky. Generating 2D rotations, translations and scalings of an image by itself is trivial: we don't need a CNN for that. The reason behind training a CNN capable of these transformations might be that such models would then have an understanding of this transformation just as humans have, and then we can hope that the representations learned by these models can generalize on downstream tasks. However, this has not been explored, or sufficiently discussed in the paper currently. In a nutshell - why are we training models to rotate/translate? What kind of OOD situations should these models generalize and be able to rotate/translate, and why is that a necessary requirement?

5. Not placed in the context of existing work: Several ideas discussed in the paper have been explored in recent literature which have not been mentioned/discussed. For instance, the idea of novel shape-viewpoint and the role of data diversity on generalization to OOD has been discussed at length in [1]. Similarly, generalization to OOD viewpoints has been discussed in several generative works on novel view synthesis [2,3,4]. Also, the interplay between convolutions and translation invariance has been studied in [5,6]. Placing this work in the context of existing literature in empirical analysis of invariances is essential to make its contributions clear.

I am happy to revise my review if I have misunderstood any details, or there is an error in the review. Please feel free to point them out!

References

1. Madan, S., Henry, T., Dozier, J., Ho, H., Bhandari, N., Sasaki, T., Durand, F., Pfister, H. and Boix, X., 2020. When and how do CNNs generalize to out-of-distribution category-viewpoint combinations? arXiv preprint arXiv:2007.08032.

2. Borji, A., Izadi, S. and Itti, L., 2016. ilab-20m: A large-scale controlled object dataset to investigate deep learning. In Proceedings of the IEEE Conference on Computer Vision and Pattern Recognition (pp. 2221-2230).

3. Gross, R., Matthews, I., Cohn, J., Kanade, T. and Baker, S., 2010. Multi-pie. Image and vision computing, 28(5), pp.807-813.

4. Qiu, W., Zhong, F., Zhang, Y., Qiao, S., Xiao, Z., Kim, T.S. and Wang, Y., 2017, October. Unrealcv: Virtual worlds for computer vision. In Proceedings of the 25th ACM international conference on Multimedia (pp. 1221-1224).

5. Zhang, R., 2019, May. Making convolutional networks shift-invariant again. In International conference on machine learning (pp. 7324-7334). PMLR.

6. Chaman, A. and Dokmanic, I., 2021. Truly shift-invariant convolutional neural networks. In Proceedings of the IEEE/CVF Conference on Computer Vision and Pattern Recognition (pp. 3773-3783).


**Summary Of The Paper:**

At its heart, the paper explores which inductive biases (and image representations) enable CNNs to generalize better across 2D transformations including rotations, translations and scale variations. The paper starts by proposing a methodology for generating a controlled dataset of binary masks consisting of random polygons. All training and testing is conducted on this dataset. After showing that convolutions and high data diversity enable better generalization across these transformations, the paper proposes two additional ways to improve performance. Firstly, their approach transforms input images into log-polar space, as object rotations in the original space is equivalent to translations in this space, and thus convolutions on log-polar space are equivariant to rotations in the original pixel space. Secondly, the paper proposes an iterative approach to model transformations, where output mask generated by their Encoder-Decoder model is fed back into the network as an input. The motivation behind this iterative approach is that these transformations form a symmetric group, and so the output after k iterations is still a member of the set which can be reached via a single transformation. The motivation is that such training would force the network to preserve the shape being transformed across iterations. The paper shows that networks trained with these inductive biases and training methodologies do indeed generalize better across these 2D transformations.

**Summary Of The Review:**

While the paper proposes some interesting ideas (iteration, log-polar convolutions) and attempts to understand generalization across 2D transformations in a rigorous manner, it is currently not mature enough for publication. I would advise (1) extending the experiments on some more realistic datasets listed below, (2) paying greater attention to the training/testing distributions used, and (3) placing the work in the context of existing works above all to make this work more solid.

---

### Official Review · Reviewer_31wD · 2021-11-03

**Correctness:** 4
**Technical Novelty And Significance:** 3
**Empirical Novelty And Significance:** 3
**Recommendation:** 6
**Confidence:** 3

**Main Review:**

Strength:

- This authors use a unique setup to test the generalization capabilities of models. In particular, their data generation strategy is flexible and can produce a large variety of images with different rotations, scales, and translations.

- The authors test the ability of the models to extrapolate beyond the transformations present in training, and identify 4 ways to improve generalization ability (iterative training, diversity, convolutions, log-polar coordinates)

-The authors show that some strategies are effective for translation, but more is needed to address rotation and scaling.

-The paper is well written and interesting to read.

Weaknesses:

-Though the data generation setup and test environment is interesting and unique, it's not clear how effective the techniques will be on more complicated realistic datasets. We already know that diversity and convolutions are useful tools. There's also prior work showing iterative training is useful for addressing domain adaptation https://arxiv.org/abs/2002.11361. So, the novel recommendation is to try log-polar coordinates. It would be interesting to test if log-polar coordinates can help OOD generalization on ImageNet.

**Summary Of The Paper:**

The authors trained autoencoders and variational autoencoders in cartesian space, as well as autoencoders in log-polar space, with generated data representing a range of interesting transformations. They then tested the ability of the models to extrapolate beyond the learned transformations in pixel space. The authors find that iterative training, data diversity, convolutions, and transformation to log-polar space all improve the generalization performance.


**Summary Of The Review:**

The paper provides a unique way to test generalization abilities of models and offers some novel recommendations and insights.

My score reflects my view that the data generation process and test environment that the authors propose are important and novel contributions.

---

### Decision · Program_Chairs · 2022-01-20

**Decision:**

Reject

**Comment:**

This paper studies different inductive biases that would improve OOD generalization (and in particular under translation, rotation and scaling) for image tasks. The study is focused on a toy dataset which allows authors to have more control over the data generation process and the transformations. Authors further show that iterative training using an auto-encoder and presenting data in log-polar space helps with rotation and scaling transformations on their toy dataset.


Strong Points:
- The paper is well written and easy to follow.
- The data generation process and the resulting toy dataset are novel and interesting.
- The experiment design and evaluations are solid.

Weak Points:
- No natural image datasets: While using a toy dataset has several benefits it does not grant that the conclusions would generalize to realistic settings. Reviewers have suggested several realistic datasets and I encourage authors to evaluate their findings on some of these datasets.
- Limited Baselines: As reviewers have pointed, comparison with baselines can be improved by including stronger baselines as well as more clear discussion about other techniques such as augmenting the data with the transformations.
- Related Work: A proper discussion of related work to set the context and highlight the contributions of this paper is missing. In particular, reviewers have pointed to prior work on the benefits of presenting the image in log-polar space.

Unfortunately, authors did not engage with reviewers during the discussion period. Given the prior work and lack of any natural image dataset, I think the novelty and significance of this work is limited. Therefore, I recommend rejecting the paper. However, I encourage authors to improve the paper by addressing the points raised by the reviewers.